# A Robustness Analysis of a Fuzzy Fractional Order PID Controller Based on Genetic Algorithm for a DC-DC Boost Converter

**Luís Felipe da S. C. Pereira** , **Edson Batista \*** , **Moacyr A. G. de Brito** and **Ruben B. Godoy**

Faculty of Engineering, Architecture and Urban Planning and Geography,
Federal University of Mato Grosso do Sul, Campo Grande 79070-900, Brazil; luis.pereira@ufms.br (L.F.d.S.C.P.);
moacyr.brito@ufms.br (M.A.G.d.B.); ruben.godoy@ufms.br (R.B.G.)
**\*** Correspondence: edson.batista@ufms.br

**Abstract:** In this paper, a new topology of a Fractional Order PID (FOPID) controller is proposed to control a boost DC-DC converter with minimum over/undershoot. The fractional controller parameters are tuned using a genetic algorithm (GA) with a combined cost function composed of the Integral of Time-Weighted Absolute Error (ITAE) and the Integral of Time-Weighted Square Error (ITSE). Despite adding moderate complexity to the control structure, the simulation results reveal that the GA-based FOPID controller tuning provided better performance for the setpoint tracking both under load variations and parameters deviation due to the prolonged use. The proposed FOPID shows a wide operational range concerning load disturbances, and capacitance/inductance deviations of ±30% and ±50% from nominal values, achieving functionality and voltage stability even with output power 50% higher than the converter power specification. The assessment was made considering operation in voltage mode and the performance was compared to conventional Proportional-Integral (PI), Type II and current mode controllers. Finally, a fuzzy fractional-order PID (FFOPID) was designed to outperform the FOPID during disturbances in the control variable.

**Keywords:** boost DC-DC converter; fractional order PID controller; fuzzy logic controller; genetic algorithm

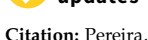



## 1. Introduction

Since the middle of the last century, proportional-integral-derivative (PID) controllers have been used in the control of many industrial processes. During the 1940s, Ziegler and Nichols [1] launched the famous method for tuning PID controllers based on the transient response characteristic of a given plant. Following them, many engineers and scientists developed adjustment methods applied to the synthesis of PID controllers [2–5].

The fractional calculus is also a well-known theory and can solve a lack of answers about integer calculus [6]. However, due to the extremely high computational cost, it has been neglected for digital implementations. Nonetheless, nowadays, hardware constraints and computing costs are no more preventive issues for embedded systems [7]. Thus, applications using fractional order controllers have become doable and lines of research focused on control optimization are feasible for non-integer degrees. Recently, many works have been published with tuning Fractional Order PID (FOPID) techniques in different subjects exploring the higher range of the fractional parameters [8–13].

As regards [14], a FOPID controller based on Queen Bee Assisted Genetic Algorithm (QBGA) was designed to outperform the PID controller when parameter variations are applied. Again, the superiority of the FOPID is remarkable; nevertheless, the paper makes no study related to load disturbances and capacitance variations. In addition, no optimized search range of parameters was introduced to the algorithm. Ref. [15] compared the start-up performance and step response of optimal PID and optimal FOPID controllers applied to the non-linear boost converter with the Artificial Bee Colony (ABC) algorithm as the

tuning method to find the FOPID parameters. The non-linear boost converter feature of [15] was designed by the SIMULINK Simscape tool and the Bee Colony algorithm ran online to achieve sub-optimal parameters, but no robustness evaluations were made regarding parameter deviation. Ref. [16] designed and implemented a digital FOPID controller applied to a linearized boost converter, such that a procedure to design the parameters of a FOPID controller was given together with a discretized DSP-based control algorithm and tested under different load conditions. Despite being an interesting approach, the FOPID proposed in [16] utilized two sensors, i.e., voltage and current sensors, along with a non-optimized population range, increasing the computational burden and making hardware synthesis more expensive. Moreover, it was also recommended to use $0 \leq \mu, \lambda \leq 1$, with $\mu << 1$ and $\lambda \approx 1$ to guarantee closed-loop stability, showing a small range of optimal solutions for the proposed method. The approximation of a non-integer order PID-type controller to regulate the output voltage of a DC-DC Boost converter is proposed in [17]. The Laplacian operator biquadratic approximation was utilized such that a flat phase response in a range around a center frequency is obtained, aiming at the iso-damping response of the controlled system. Experimental results were also presented to assert the good performance and regulation for the non-minimum phase Boost system. However, again, no study related to load disturbances and parameter variations was performed. In our work, a comprehensive analysis of all parameters variation and load disturbance applied to a DC-DC Boost converter controlled by the FOPID controller is presented. None of the works in the related literature explored such a complete study as well as included zero over/undershoot under closed-loop performance.

Among the optimization algorithms, the Genetic Algorithm (GA) is commonly used to generate high-quality solutions for optimization problems within a well-known range of parameters whose small amount of system information is enough to find large solutions space [18]. By using GA, this manuscript assays the FOPID controller synthesis of a fixed-frequency DC-DC Boost converter with no necessity for complex equations. The control intends to drive the power switch with a duty cycle in such a way that the output voltage reproduces the desired nominal voltage. This tracking needs to be sustained even under input voltage disturbances and load variations, keeping the project requirements during transients. The more common control techniques used in practice are based on PI-type controllers that are tuned on the basis of linearized averaged models employing voltage or current control in either one or two-loop control [19]. However, those approaches provide deficient overshoot voltage responses and poor settling times for load disturbance situations [20]. Here, the proposed FOPID based on GA can achieve a lower overshoot (lower than 1%) with a suitable settling time owing to the wider fractional parameters range accepted by the new proposed topology ($0 < \lambda, \mu < 2$).

Fuzzy logic resembles the human behavior of decision by using a set of linguistic variables denoted by membership functions and their shapes. According to fuzzy implication functions and an inference system, a fuzzy logic controller (FLC) can be designed when a set of inference rules are created to translate some action and its rate in fuzzy linguistic variables and "defuzzificate" them to obtain a crisp solution. This human behavior feature of the FLC exhibited success to control nonlinear and linear systems [21]. In this manuscript, GA ran to achieve optimal parameters that are the output membership functions center of the universe of discourse. One can say that a huge contribution of the FLC is to adapt the system when it is under disturbances. Ref. [22] designed an FLC to use the system error and derivative of the error inputs to obtain the scaling factor of the proportional, integral, and derivative terms of a predefined FOPID controller during its operation. The adaptive method provided by the fuzzy system improved the dynamic performances of the FOPID controller through which the controller may respond quickly to disturbances upon the control variable, but the FOPID design was made with the FOMCON toolbox and without any optimized population of initial FOPID parameters. To enhance all FOPID achievements, a fuzzy FOPID is designed with better robustness against parameters variation and load disturbances in an operational range from almost no power to 150% full power, for the voltage control of the DC-DC Boost converter, thus fulfilling the wider functionality

of electronic designs. It is important to highlight that the proposed controller topology achieved such outstanding performance with only one voltage sensor. More works related to fuzzy strategy to mitigate real-time disturbances can be found in [23,24].

This paper proposes the application of a genetic algorithm (GA) in a new FOPID topology that overcomes the startup undershoot problem in the voltage output of a DC-DC Boost converter. Furthermore, a fuzzy logic controller was designed with gain parameters of the FOPID-based GA as the center of the universe of discourse to tune online the gain parameters of the proposed Fuzzy FOPID (FFOPID). The proposed approach revealed superior robustness in comparison with traditional controllers. Based on those facts, the achievements of this paper are:

- The insertion of a FOPID into the closed-loop control of a DC-DC Boost converter to improve the robustness against capacitance and inductance deviations when the load resistance is changed during operation, without the insertion of high complexities in the controller synthesis;
- A new FOPID topology that overcomes the over/undershoot problem of the voltage-loop DC-DC Boost by guaranteeing the closed-loop system with initial zero derivative;
- A fuzzy logic controller is used to self-tuning the gain parameters of the FOPID to enhance its controllability as related to disturbance injection.
- Finally, comparisons among the proposed fuzzy logic FOPID controller with several conventional controllers were performed, such as PI controller, type II compensator and current mode controller. In all cases, the proposed controller outperformed those controllers regarding load disturbances and parameters variations.

This manuscript is organized as follows. The materials and methods are outlined in Section 2 by introducing the definition of the Oustaloup Filter, basic concepts of fractional calculus, and the DC-DC Boost converter plant. Further, the control approach is proposed for the DC-DC Boost converter model along with the GA implementation and its operation regarding Integral of Time-Weighted Absolute Error (ITAE) and Integral of Time-Weighted Square Error (ITSE) indexes optimizations as cost functions in this section as well. An FLC is also designed for online updating the FOPID gain parameters. Section 3 includes simulations of the DC-DC Boost converter tuned with the FOPID controller for steps responses and regulatory cases of load, capacitance, and inductance, comparing its robustness with other acquaintance controllers. Moreover, the undershoot rejection is presented in this section as well. In addition, enhancement of the fuzzy fractional-order PID (FFOPID) is shown in comparison with FOPID performance indexes under steps on control variables. A comparative discussion is developed in Section 4. Finally, our final remarks are stated in Section 5.

## 2. Materials and Methods

### 2.1. Oustaloup Filter

The main problem of fractional calculus regarding the transfer function implementation in software is its irrational and non-finite feature. Here, the Oustaloup recursive filter is employed to fit the fractional-order operators $s^\mu$ in a multiplication of 2N+1 rational functions on integer-order to achieve a good resolution in a range defined by $[\omega_b, \omega_h]$ [25,26]. The filter is defined as [26]:

$$G(s) = K \prod_{k=-N}^{N} \frac{s + \omega_k^{'}}{s + \omega_k},\tag{1}$$

where the poles, zeros, and gain of the filter (1) can be evaluated according to (2)–(4):

$$\omega_k^{'} = \omega_b \left(\frac{\omega_h}{\omega_b}\right)^{\dfrac{k + N + \frac{1}{2}(1-\gamma)}{2N+1}},\tag{2}$$

$$\omega_k = \omega_b \left(\frac{\omega_h}{\omega_b}\right)^{\dfrac{k + N + \frac{1}{2}(1+\gamma)}{2N+1}}, \tag{3}$$

$$K = \omega_h^{\gamma}. \tag{4}$$

In MATLAB, the function ousta_fod() [27] was created to generate the rational transfer function (1). Furthermore, to design the fractional operator in SIMULINK, it is necessary to insert a first-order low-pass filter with crossover frequency $\omega_h$ together with the ousta_fod() block to avoid algebraic loops [26].

### 2.2. Basic Concepts of Fractional Calculus

Fractional calculus has arisen from the integer order extrapolation to real order in differential and integral operators. In this manuscript, the Riemann–Liouville fractional integral and the Caputo fractional differentiator with their Laplace transforms will be defined.

The Riemann–Liouville fractional integral [28] of order $\lambda \in \mathbb{R}_+$ is defined as:

$$_{RL}D^{-\lambda}x(t) = \frac{1}{\Gamma(\lambda)} \int_0^t (t-s)^{\lambda-1} x(s)\, ds, \tag{5}$$

for $(n - 1 < \lambda \leq n)$ where $n \in \mathbb{N}_+$.

The Caputo fractional derivative [29] of order $\mu \in \mathbb{R}_+$ is defined as:

$$_{C}D^{\mu}x(t) = \frac{1}{\Gamma(n-\mu)} \int_0^t \frac{(x)^{(n)}(s)}{(t-s)^{\mu+1-n}}\, ds, \tag{6}$$

for $(n - 1 < \mu \leq n)$ where $n \in \mathbb{N}_+$.

The Laplace transform of the Riemann–Liouville Integral [30] is given as:

$$L[_{RL}D^{-\lambda}x(t)] = s^{-\lambda}X(s). \tag{7}$$

The Laplace transform of the Caputo Derivative [30] is given as:

$$L[_{C}D^{\mu}x(t)] = s^{\mu}X(s) + \sum_{k=0}^{n-1} s^{\mu-k-1}x^{(k)}(0), \tag{8}$$

for $(n - 1 < \mu \leq n)$, $n \in \mathbb{N}_+$. Note that (8) is performed with the integer $k$-th order derivatives of the initial $x(t)$, i.e., $x^k(0)$, this physical feature will be explored further to achieve the feasibility of the proposed new topology. Note also that $s^{-\lambda}$ and $s^{\mu}$ from (7) and (8) can be easily replaced by the Oustaloup recursive filter.

### 2.3. DC-DC Boost Description

A DC-DC Boost converter is a switch-mode regulator that steps up the input voltage to deliver a higher output voltage to feed loads or a constant DC bus. In this manuscript, the boost converter circuitry is depicted in Figure 1.

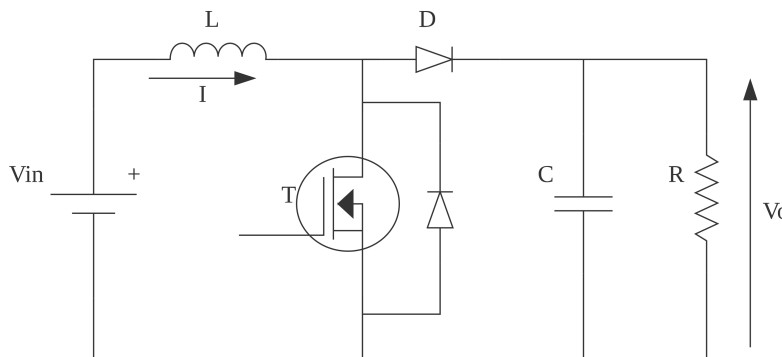

**Figure 1.** DC-DC Boost Converter Circuit.

The boost converter features two stages of operation with two different dynamics. Here, continuous conduction mode (CCM) is considered [31]. The controller modifies the control input, the duty cycle $d(t)$, for each time $t$, determining when the transistor T is conducting or not. During the time interval $\dfrac{D}{f_{sw}}$, T conducts so the inductor is charged. At the next period, T stops conducting thus power is transferred to the load through Diode D. Then, at the end of this period, T becomes enabled again. Thus, the converter operates in CCM. The converter specification and its components parameters are given in Tables 1 and 2.

**Table 1.** Specifications of converter prototype.

| Specification | Value |
|---|---|
| Input Voltage, Vin | 50 V |
| Output Voltage, Vo | 200 V |
| Duty Cycle, D | 0.75 |
| Switching Frequency, $f_{sw}$ | 20 kHz |
| Output power, Po | 400 W |
| Inductor current, I | 8 A |
| Maximum current ripple in %, | 10% |
| Maximum voltage ripple in %, | 1% |

**Table 2.** Components parameters.

| Specification | Value |
|---|---|
| Load Resistance, R | 100 Ω |
| Inductance, L | 2.34 mH |
| Output Capacitance, C | 37.5 μF |

By state-space averaging modelling [32], the transfer function for a voltage-mode boost converter in CCM can be obtained as:

$$\frac{\tilde{v}_o(s)}{\tilde{d}(s)} = \frac{V_o(1-D) - sLI}{LCs^2 + \dfrac{L}{R}s + (1-D)^2}, \tag{9}$$

where $\tilde{v}_o(s)$ and $\tilde{d}(s)$ are small-signal perturbations about the average steady-state values of *Vo* and $D$.

Equation (9) is a non-minimum phase plant with undesirable over/undershoot when disturbed by steps. Ref. [32] describes a method employing two-control loops, one for the inductor current and other for the output voltage to reduce the influences of the non-minimum phase effects. This approach uses two sensors. Another approach considers the usage of a very small inductance, but it increases the output ripples and the conduction losses. Here, the main idea is to improve the controller response using only the voltage

sensor, which is less expensive and less noise-sensitive. In that sense, the control problem is reformulated into a voltage reference scheme using a FOPID controller that almost vanishes the over/undershoot and, due to resistance disturbance minimization, it has a lower voltage disturbance and less settling time.

### 2.4. Proposed Control Approach

Firstly, GA is performed to minimize the value of the objective function formed by the error between the set-point and the step reference, generating the optimal parameters of the FOPID controller for a linearized DC-DC Boost converter. Thereafter, in SIMULINK, the system ran again under several scenarios to evaluate the closed-loop system response generated by the optimal parameters. For this purpose, Figure 2 inferred that the objective of the work is to tune the FOPID parameters with minimum Integrated Time Absolute Error (ITAE) and Integrated Time Squared Error (ITSE) weighed by a probabilistic GA variable. In this model $e(t)$ is the difference between the desired output $r(t)$ and actual output $y(t)$, $d(t)$ is the FOPID control law, and parameters $K_p$, $K_i$, $K_d$ are the proportional, integral, and derivative gains, $\mu$ is the fractional derivative order, $\lambda$ is the fractional integral order and p is the probability between the Integrated Time Absolute Error (ITAE) and the Integrated Time Squared Error (ITSE).

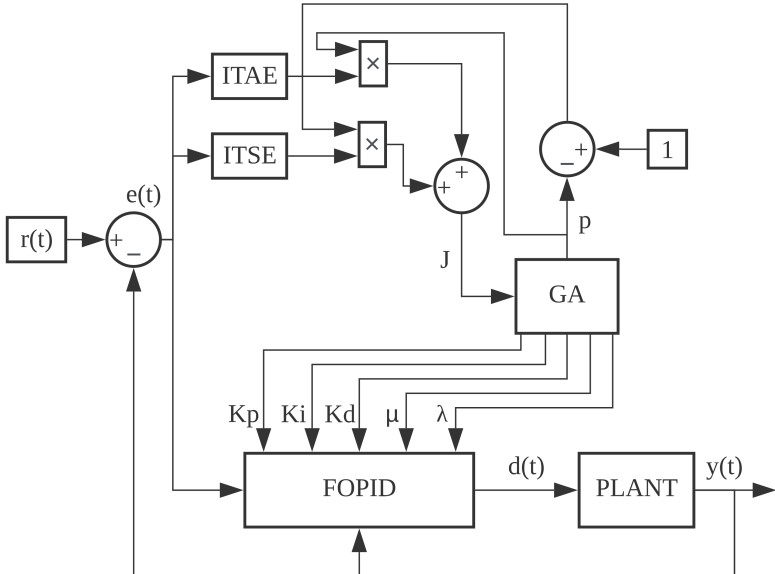

**Figure 2.** Closed Loop FOPID Controller.

The ITAE and ITSE, described as given in (10) and (11), are commonly employed error criteria to achieve optimized PID tuning values when pursuing small overshoots and short oscillation in signal accommodation [33]. From Figure 2, p creates the weighted combination of ITAE and ITSE through GA. Thus:

$$ITAE = \int_0^\infty t|e(t)|\,dx, \tag{10}$$

$$ITSE = \int_0^\infty te^2(t)\,dx. \tag{11}$$

The objective function $J$ used in GA is defined as

$$J(\vec{x}) = p \cdot ITAE + (1-p) \cdot ITSE, \tag{12}$$

where $\vec{x} = (K_p, K_i, K_d, \mu, \lambda, p)$.

### 2.5. FOPID Model Approach

As the DC-DC Boost converter, represented by the transfer function (9), is a non-minimum phase system, this manuscript uses a model to decrease undershooting when the main device is turned on, changing the place of the derivative portion of the controller to avoid the initial impulse when the reference is activated. Lemmas 1 and 2 demonstrate that the signal control and its derivative are zero at $t = 0$. The chosen controller is given in Figure 3.

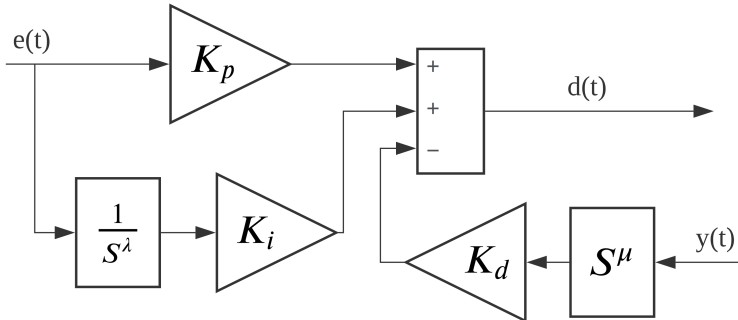

**Figure 3.** FOPID Model Controller.

In Figure 2, we assumed the output $y(t)$ and your derivative defined, besides that, $y(t)$ is a proper fraction and stable, checked by the MATLAB's isstable() function. So the initial value theorem and final value theorem are valid [34]. Thus, it can be shown this sort of controller can make the closed-loop system stable for values $0 \leq \mu \leq 2$ and $0 \leq \lambda \leq 2$ without needing of change in the proposed structure.

Thus, for $i > k$, without loss of generality in frequency domain, the plant can be described by:

$$G_p(s) = K \frac{\prod_k (s - z_k)}{\prod_i (s - p_i)}. \tag{13}$$

The closed-loop function of Figure 2 can be calculated as:

$$G_p(s) = Y(s)$$
$$\therefore Y(s)(1 + G_p(s)K_d s^\mu) = E(s)G_p(s)(K_p + K_i s^{-\lambda}). \tag{14}$$

Replacing the system error function defined by $E(s) = R(s) - Y(s)$ in (14), yields:

$$Y(s)(1 + G_p(s)K_d s^\mu) = (R(s) - Y(s))(K_p + K_i s^{-\lambda})$$
$$\therefore Y(s)[1 + G_p(s)(K_p + K_i s^{-\lambda} + K_d s^\mu)] = R(s)G_p(s)(K_p + K_i s^{-\lambda}). \tag{15}$$

Thus, the closed-loop transfer function of the system is:

$$H(s) = \frac{Y(s)}{R(s)} = \frac{G_p(s)(K_p + K_i s^{-\lambda})}{1 + G_p(s)(K_p + K_i s^{-\lambda} + K_d s^\mu)}. \tag{16}$$

Furthermore, the system error function of Figure 2 is:

$$\frac{E(s)}{R(s)} = \frac{1 + G_p(s)K_d s^\mu}{1 + G_p(s)(K_p + K_i s^{-\lambda} + K_d s^\mu)}. \tag{17}$$

**Lemma 1.** *For all $\mu, \lambda \in \mathbb{R}_+^*$, together with the later assumptions and considering the step input, we have $\lim_{t \to 0} y(t) = 0$.*

**Proof.** Using the initial value theorem in (16) and letting the higher degree of s in evidence:

$$\lim_{s\to\infty} sY(s) = \lim_{s\to\infty} \frac{Ks^k \prod_k(1-z_k/s)(K_p + k_i s^{-\lambda})}{s^i \prod_i(1-p_i/s) + Ks^{\mu+k} \prod_k(1-z_k/s)[K_d + k_p s^{-\mu} + K_i s^{-\mu-\lambda}]}. \tag{18}$$

It is straightforward to see:

$$\lim_{s\to\infty} max\{\frac{1}{s^{i-k}}; \frac{1}{s^\mu}\} = 0. \tag{19}$$

□

Despite that result, the Plant model has been assumed to have $y^{(1)}(0) = 0$. However, Equation (8) creates a derivative problem, indeed, we need to know the derivative values of $y(t)$, for $t \to 0$, when the derivative order is higher than 1. In other words, we need to find the condition that ensures $y^{(1)}(0) = 0$ because $\{\mu, \lambda\} \in [0, 2]$.

**Lemma 2.** *Considering $L(\frac{dy}{dt})$ a proper fraction and stable. Furthermore, assuming the later assumptions of the model Plant and considering the step input. If $\mu > 1$ or $i > k + 1$, thus:*

$$\lim_{t\to 0} \frac{dy}{dt} = 0. \tag{20}$$

**Proof.** Let us start with the Laplace Transform of $\frac{dy}{dt}$. It is straightforward by the first Lemma:

$$L(\frac{dy}{dt}) = sL(y) + y(0) = sY(s). \tag{21}$$

So, by the initial value theorem in (16):

$$\lim_{t\to 0} \frac{dy}{dt} = \lim_{s\to\infty} s^2 Y(s) = \lim_{s\to\infty} \frac{Ks^{k+1} \prod_k(1-z_k/s)(K_p + k_i s^{-\lambda})}{s^i \prod_i(1-p_i/s) + Ks^{\mu+k} \prod_k(1-z_k/s)[K_d + k_p s^{-\mu} + K_i s^{-\mu-\lambda}]}. \tag{22}$$

It is easy to see that (22) is equal to zero when $\mu > 1$ or $i > k + 1$. □

Thus, according to $\mu$ chosen $\in [0, 2]$ and the assumptions made about the closed-loop response, we can realize, by Lemmas 1 and 2 in (8), that:

- For $0 < \mu \leq 1$:
$$L[_C D^\mu y(t)] = s^\mu Y(s) - s^{\mu-1} y(0) = s^\mu Y(s). \tag{23}$$

- For $1 < \mu \leq 2$

$$L[_C D^\mu y(t)] = s^\mu Y(s) - s^{\mu-1} y(0) - s^{\mu-2} \frac{dy(0)}{dt} = s^\mu Y(s). \tag{24}$$

Thus, Equations (23) and (24) ensure the fractional model in Figure 3, regard as the behavior of (9) with $i - k = 1$, be possible. Comparing with the conventional parallel model [35], the closed-loop equation would be:

$$H(s)_{par} = \frac{Y(s)}{R(s)} = \frac{G_p(s)(K_p + K_i s^{-\lambda} + K_d s^\mu)}{1 + G_p(s)(K_p + K_i s^{-\lambda} + K_d s^\mu)}. \tag{25}$$

However, using the same considerations about the stability and causality of $y(t)$ as used in the demonstration of Lemmas 1 and 2, it is remarkable that $y(0)$ and $y^{(1)}(0)$ are not zero. By the same steps, the answer differs only by an order addition of $\mu$ in the numerator of (18) and (22), resulting in $y(0) = 0$ only for $\mu < 1$ and $y^{(1)}(0) \neq 0$ for all $\mu$. Those differences will be realized over the initial undershooting voltage analysis among the controllers described in (16) and (25) where the plant in (16) presents better answer as seen in Section 4.

### 2.5.1. Genetic Algorithm Description

A genetic algorithm is a metaheuristic algorithm based on nature life generation. According to an initial population of guesses, identified as chromosomes, the algorithm follows several nature rules toward an optimal solution. The chromosomes held in this manuscript are created with base-10 encoding and generated with a continuous random function. One can say that GA has a high diversity feature and so it can handle problems with a wide number of variables. A typical GA possesses three key genetic operators such as selection, crossover, and mutation [36]. Figure 4 describes the GA flowchart diagram used in this manuscript with a linear constraint applied in GA between the ITAE and ITSE indexes over the objective function J. Table 3 depicts the parameters used in GA.

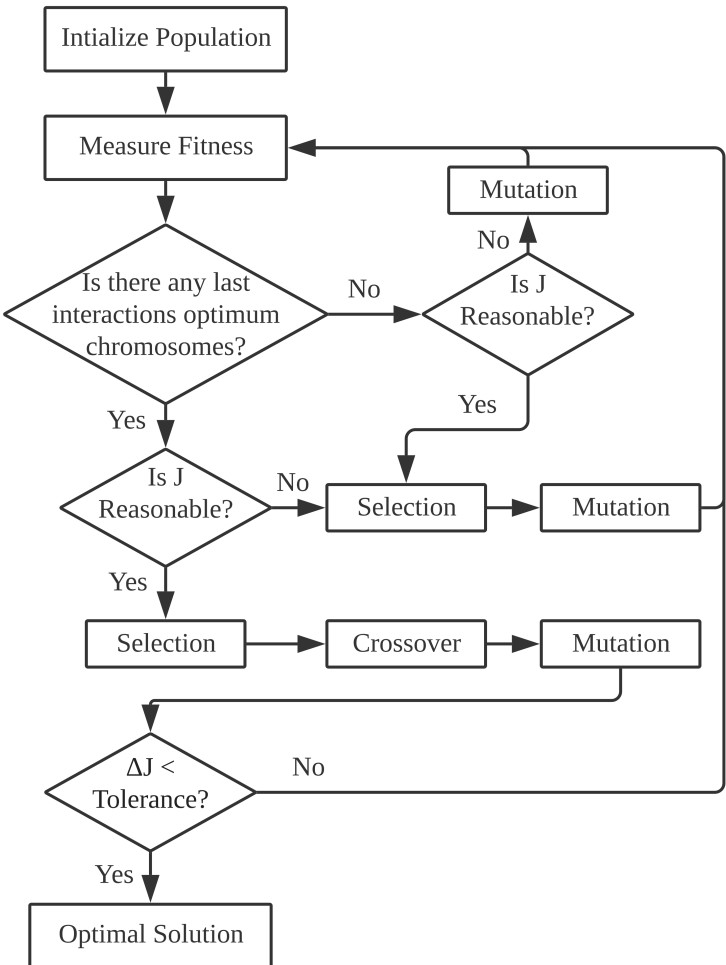

**Figure 4.** Flowchart Diagram of GA.

**Table 3.** Parameters used in GA.

| Parameter | Value/type |
| --- | --- |
| Selection | Steady State |
| Maximum generations | 50 |
| Population size | 10 |
| Crossover | Single point crossover |
| Mutation | Uniform distribution |

The fitness, reproduction, crossover, and mutation steps of the GA is described, briefly, as follows:

**Fitness:** To improve the fitness, chromosomes are the start point for the unconstrained Nelder–Mead Simplex algorithm [37] implemented in MATLAB's Optimization Toolbox function fminsearch. The objective function (12) is the function to be minimize for every chromosome generated whether by initialization or mutation with lower and upper bounds of the FOPID parameters.

**Reproduction:** Reproduction is a basic operator of convergence in GA due to its survival selection mechanism. In this manuscript, the selection type chosen was the steady-state with at least 50% of the best possible chromosomes. A distinction is also made between the acceptable chromosome by probabilistic measurement, so the next generation is reproduced with the last fittest chromosomes as the mean value of the uniform distribution over the closest parameters edge. Regarding rule (12), the best chromosomes have lower errors; thus, the probability is inverted among the errors to use them more in reproduction stage. Thus, the reproducibility P by each fittest chromosome is defined as:

$$P = \frac{\frac{1}{p_e}}{\sum_{i=1}^{n_e} \frac{1}{p_e}}, \tag{26}$$

where $n_e$ is the number of the fittest chromosomes and $p_e$ the probability over their errors. According to this reproducibility, the remaining non-selected part is replaced by mutation from the P selected chromosomes.

**Crossover:** Crossover is an operator to improve diversity among the set of the best chromosomes. The technique reflects the natural exchange information of sexual reproduction between natural organisms [38]. In this manuscript, the single-point crossover method is used [39] over the fittest chromosomes. It is worth mentioning that the fitness operator is performed after the crossover operator to evaluate the new solutions.

**Mutation:** Unfortunately, while the GA runs, the exchange of genes among chromosomes starts to be lower due to the dominating of the fittest chromosomes. Consequently, after several generations the non-mutation leads to premature convergence of nonoptimal solution. To overcome this undesirable issue, a random change is used by uniform distribution over the remaining chromosomes to complete the population size.

According to Figure 4, the GA based FOPID tuning is better summarized as follows:

**Step 1:** GA starts by a random population over uniform distribution. The parameters range is displayed in Table 4 and the range over parameters $K_p, K_i, K_d$ has come by the integer Routh–Hurwitz stability criterion in (16), when $\mu = 1$ and $\lambda = 1$, because of the fractional range is about one.

**Step 2:** At each generation, each chromosome is designed as $(K_p, K_i, K_d, \mu, \lambda, p)$ and its fitness is calculated over the FOPID controller using the probabilistic factor $p$ to weight the cost functions of ITAE and ITSE. If any optimization happened before and the new $J$ is reasonable then selection, crossover and mutation are made. Otherwise, depending on the last optimizations and values of $J$, the algorithm selects and mutates or only mutates for the next generation.

**Step 3:** If the difference over the lower values of $J$ achieves value less than the tolerance, the algorithm finds an optimal solution, otherwise it will run itself again until the last generation.

The ultimate aim of GA is to seek global PID values $(K_p, K_i, K_d, \mu, \lambda, p)$ with minimum fitness value to operate the DC-DC Boost converter in the entire range (non-load to full load). Operating a power converter in the entire range is a challenging task, normally the converter works with a minimum load to avoid malfunctioning. This highlights the proposed FOPID achievements.

**Table 4.** GA Parameters Range.

| Parameter | $K_p$ | $\mu$ | $\lambda$ | $K_i$ | $K_d$ | p |
|---|---|---|---|---|---|---|
| Minimum value | 0 | 0.8 | 0.8 | 0 | 0 | 0 |
| Maximum value | $1.25 \times 10^{-4}$ | 1.2 | 1.2 | 6.6774 | $4.6875 \times 10^{-6}$ | 1 |

### 2.5.2. Fuzzy FOPID Controller

This manuscript uses three fuzzy inference systems with two inputs and one output for each parameter to self-tuning the FOPID controller [22]. The controller design has used the error and the change of error, either normalized with the voltage reference, as inputs and the gains $\Delta K_p$, $\Delta K_i$, $\Delta K_d$ as outputs to the self-tuning. The FLC outputs were added to the FOPID parameters proposed to adjust online the controller behavior concerning control variable disturbances. The schematic of the FFOPID control structure utilized in this manuscript is shown in Figure 5.

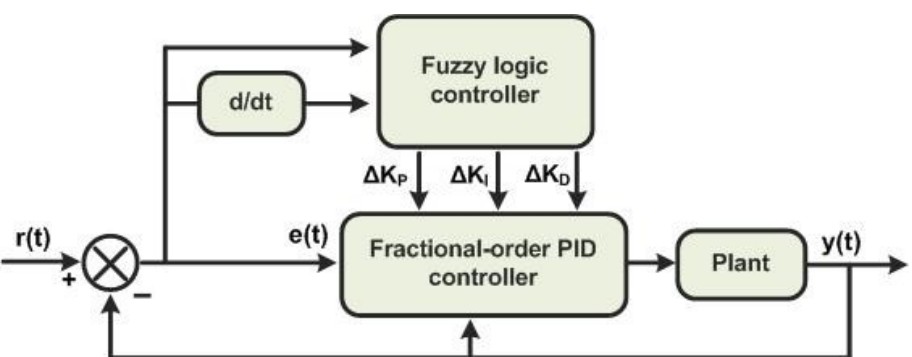

**Figure 5.** Block diagram of the FFOPID controller.

Updating the parameters, the final gain values $K_{pf}$, $K_{if}$ and $K_{df}$ of the FOPID controller are described as follows:

$$K_{pf} = K_p + \Delta K_p; \tag{27a}$$

$$K_{if} = K_i + \Delta K_i; \tag{27b}$$

$$K_{df} = K_d + \Delta K_d, \tag{27c}$$

where $K_p$, $K_i$ and $K_d$ are the initial gain value of the proposed FOPID controller and $\Delta K_p$, $\Delta K_i$, $\Delta K_d$ the scaling factors computed from FLC. For simplicity, the Mamdani-type fuzzy inference system is applied while the deviation ranges of output variables (universe of discourse) were defined as 50% of each initial gain value. For membership functions, the triangular shape was used for input and output fuzzy sets. As regards the linguistic variable names, the membership functions were denoted as NB (negative big), NS (negative small), N (negative), Z (zero), P (positive), PS (positive small), PB (positive big) and they are depicted in Figures 6 and 7. From Tables 5–7, the rule base used for each output is shown. It is remarkable that the universe of discourse from the input variables is between $-1$ and 1 due to their normalization upon the voltage reference and a digital design of the FLC with a sampling time of $10^{-4}$ s. The gain inserted before the fuzzy inference for the error and the change-in-error were 1.1 and 2, respectively. Finally, the center of gravity defuzzification method is selected to determine the crisp output.

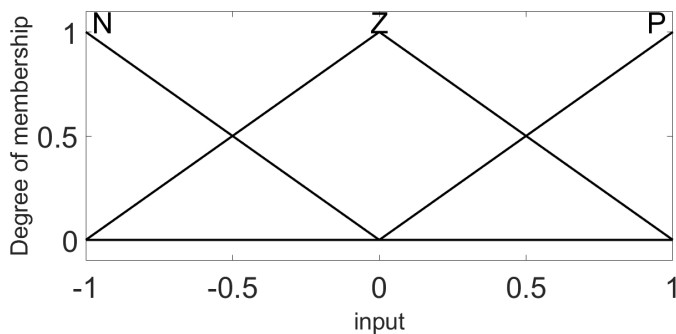

**Figure 6.** Fuzzy membership functions of input error and change-in-error.

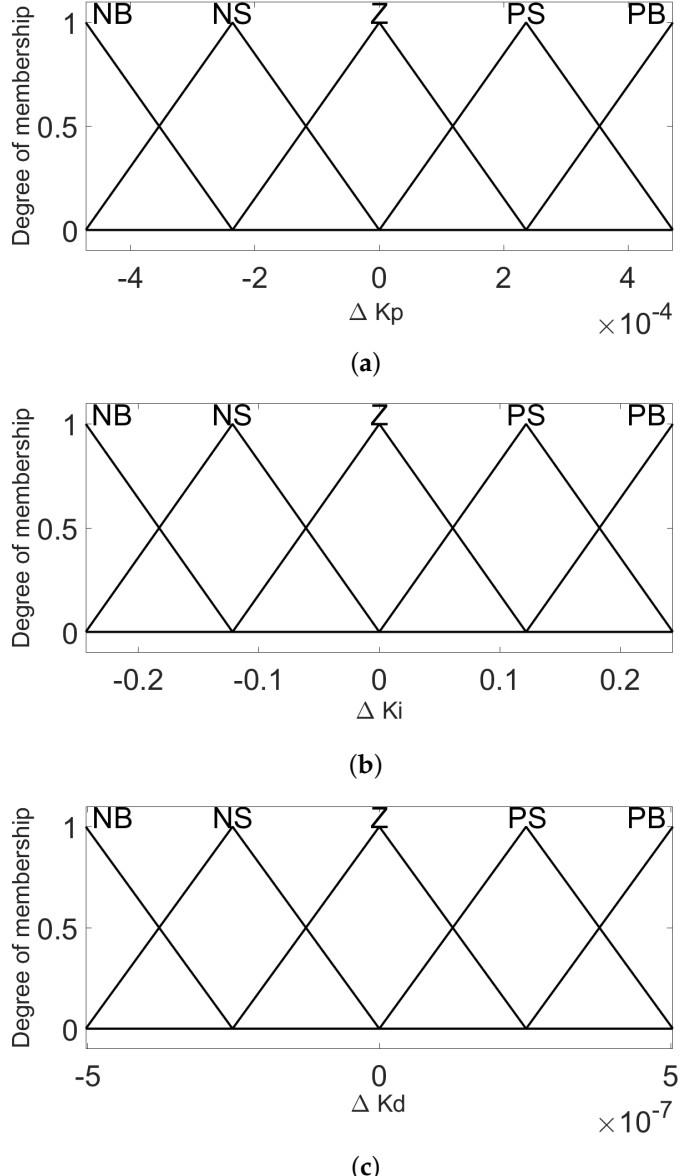

**Figure 7.** (**a**) Fuzzy membership functions of the output parameters of $\Delta K_p$. (**b**) Fuzzy membership functions of the output parameters of $\Delta K_i$. (**c**) Fuzzy membership functions of the output parameters of $\Delta K_d$.

**Table 5.** Fuzzy linguistic rule-base for $\Delta K_p$.

| $\Delta K_p$ | | e | | |
|:---:|:---:|:---:|:---:|:---:|
| | | **N** | **Z** | **P** |
| | N | NB | NS | Z |
| $\Delta e$ | Z | NB | NS | NS |
| | P | Z | PS | PS |

**Table 6.** Fuzzy linguistic rule-base for $\Delta K_i$.

| $\Delta K_i$ | | e | | |
|:---:|:---:|:---:|:---:|:---:|
| | | **N** | **Z** | **P** |
| | N | NB | NS | NS |
| $\Delta e$ | Z | NS | Z | PS |
| | P | Z | PS | PB |

**Table 7.** Fuzzy linguistic rule-base for $\Delta K_d$.

| $\Delta K_d$ | | e | | |
|:---:|:---:|:---:|:---:|:---:|
| | | **N** | **Z** | **P** |
| | N | NS | NS | Z |
| $\Delta e$ | Z | Z | Z | PS |
| | P | Z | PS | PS |

## 3. Results

In this section, simulation results demonstrating potential advantages of the proposed control methodology are presented using the SIMULINK/MATLAB 2021a platform. For comparison purposes, gain crossover frequencies ($\omega_{gc}$) and phase margins ($\phi_m$) of PI, Type II and Current Mode controllers are related in Table 8.

**Table 8.** Gain crossover frequencies and phase margins.

| Controller | | $\omega_{gc}$ **(rad/s)** | $\phi_m$ |
|:---:|:---:|:---:|:---:|
| PI | | $2\pi18.8$ | 92.9° |
| Type II | | $2\pi18.9$ | 101° |
| Current Mode | Voltage Loop | $2\pi100$ | 76° |
| | Current Loop | $2\pi1000$ | 80° |

As mentioned in the context of robustness necessity with load disturbance, the overshoot was defined to be lower than 1% and the settling time lower than 0.05 s. By doing so, PI and Type II controllers were also designed with overshoots lower than 1% and the smallest possible settling time. For the current mode controller, $\omega_{gc}$ is typically set to be $1/10 \approx 1/5$ of the switching frequency for the current loop and consequently the same for the voltage loop. Thus, aiming for robustness instead of speed, the lower $\omega_{gc} = 2\pi20,000/10 = 2\pi1000$ has been chosen along with $45° \leq \phi_m \leq 90°$ [40]. The upper limit of the integral performance indices is chosen as 0.1 s. Moreover, the Oustaloup filter order chosen was $N = 7$ with frequency resolution $[\omega_b, \omega_h] = [10^{-4}, 10^5]$ rad/s. The best solution related to load disturbance robustness among all simulations found is given in Table 9.

**Table 9.** Optimal parameters of FOPID.

| $K_p$ | $\mu$ | $\lambda$ | $K_i$ | $K_d$ | $p$ |
|:---:|:---:|:---:|:---:|:---:|:---:|
| $9.43944 \times 10^{-4}$ | 1.07965 | 1.00013 | 0.48732 | $1.00748 \times 10^{-6}$ | 0.52973 |

The first case to be analyzed is the transient behavior during startup. Applying an input voltage of 200 V, Figure 8 depicts step responses of different controllers during startup. Notice that the FFOPID controller respects the maximum voltage overshoot of 1% while the Current Mode controller reaches 5%. However, this trade-off is remarkable with the rise time.

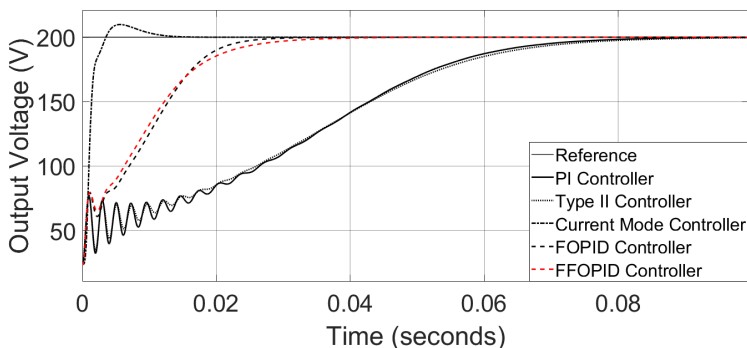

**Figure 8.** Simulation results of controllers behavior in transient time.

As regard to the second case, the load disturbance was analyzed. The resistance R was changed by a step response after 200 and 350 ms from the startup. Figure 9a shows the voltage response over the R disturbance from 100 Ω to 80 Ω and Figure 9b from 100 Ω to 66.67 Ω. It is remarkable that the FFOPID controller has the fastest stability over the load disturbance through Figure 9a–d, even with only one voltage sensor. When the load disturbance is changed towards the above, it is noticeable that the FFOPID controller is still stable for high resistances as seen in Figure 10 while the Current Mode Controller is unstable. The PI controller was not drawn due to their high instability. The changing load from 100 Ω to 10 kΩ represents the operation from full load to a practically non-load. No conventional controllers can handle such high disturbance and thus, the viability of the proposed FFOPID controller is demonstrated.

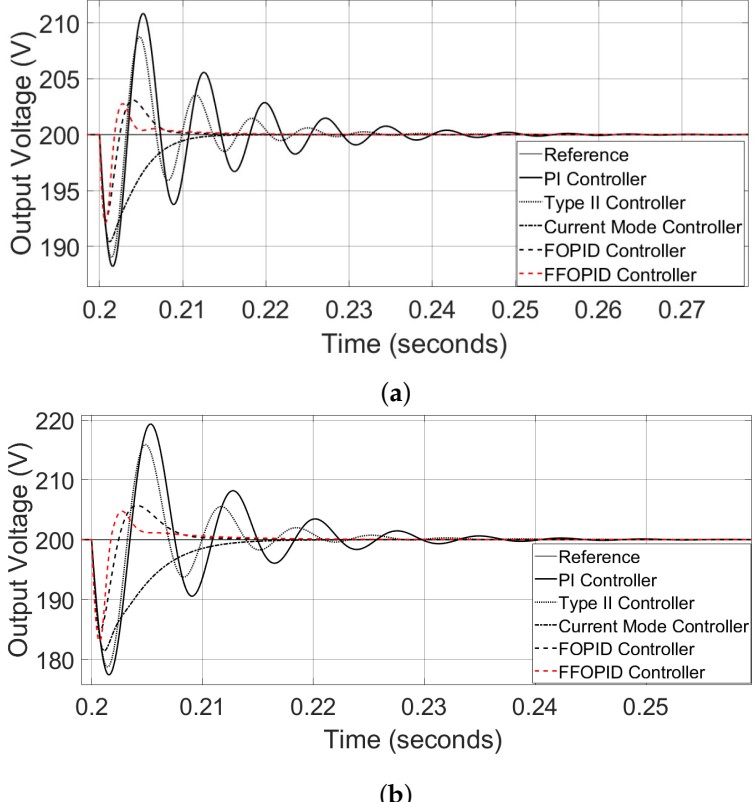

(**a**)

(**b**)

**Figure 9.** *Cont.*

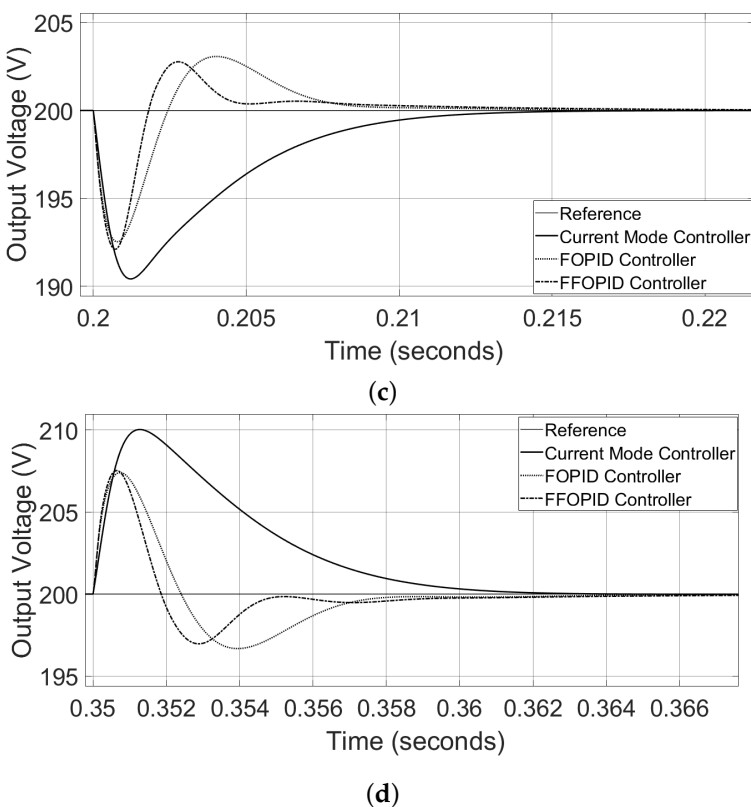

**Figure 9.** (**a**) Output voltage response over the R disturbance from 100 Ω to 80 Ω. (**b**) Output voltage response over the R disturbance from 100 Ω to 66.67 Ω. (**c**) Output voltage response over the R disturbance from 100 Ω to 80 Ω. (**d**) Output voltage response over the R disturbance from 80 Ω to 100 Ω.

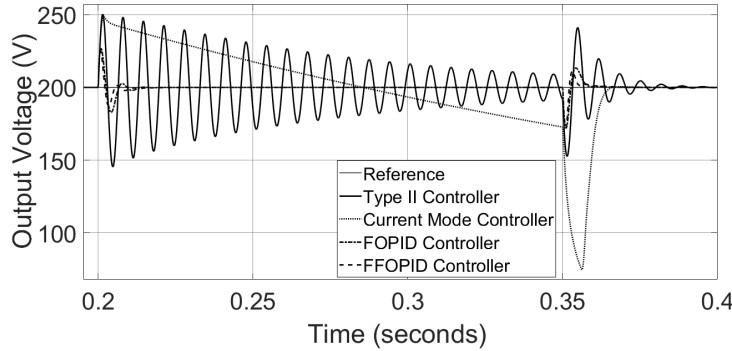

**Figure 10.** Output voltage during load disturbance from 100 Ω to 10,000 Ω and returning from 10,000 Ω to 100 Ω (full load to non-load).

The third case concerns the initial voltage behavior on the startup between the FFOPID controllers in (16) and (25). Figure 11 shows the undesirable undershoot voltage of the controller in (25). From such findings, (16) can remove the initial plant non-minimum phase effect.

The fourth case tests the robustness of the FFOPID controller against the Current Mode controller due to capacitance and inductance deviations when load resistance is switched during its operation. The startup capacitance deviation is under ±30% of its nominal value and the startup inductance is under ±50% of its nominal value. The resistance R was changed by a step response after 200 ms from the startup of 100 Ω to 80 Ω and after 350 ms from 80 Ω to 100 Ω. From Figure 12a–d, the FFOPID controller is shown to be the most robust controller against plant parameters deviations.

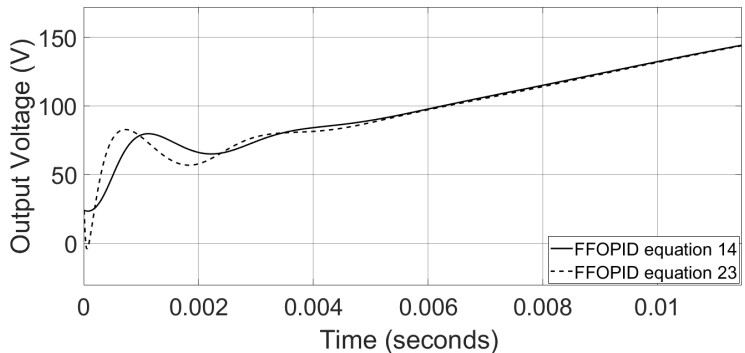

**Figure 11.** Comparative initial voltage between controllers (16) and (25).

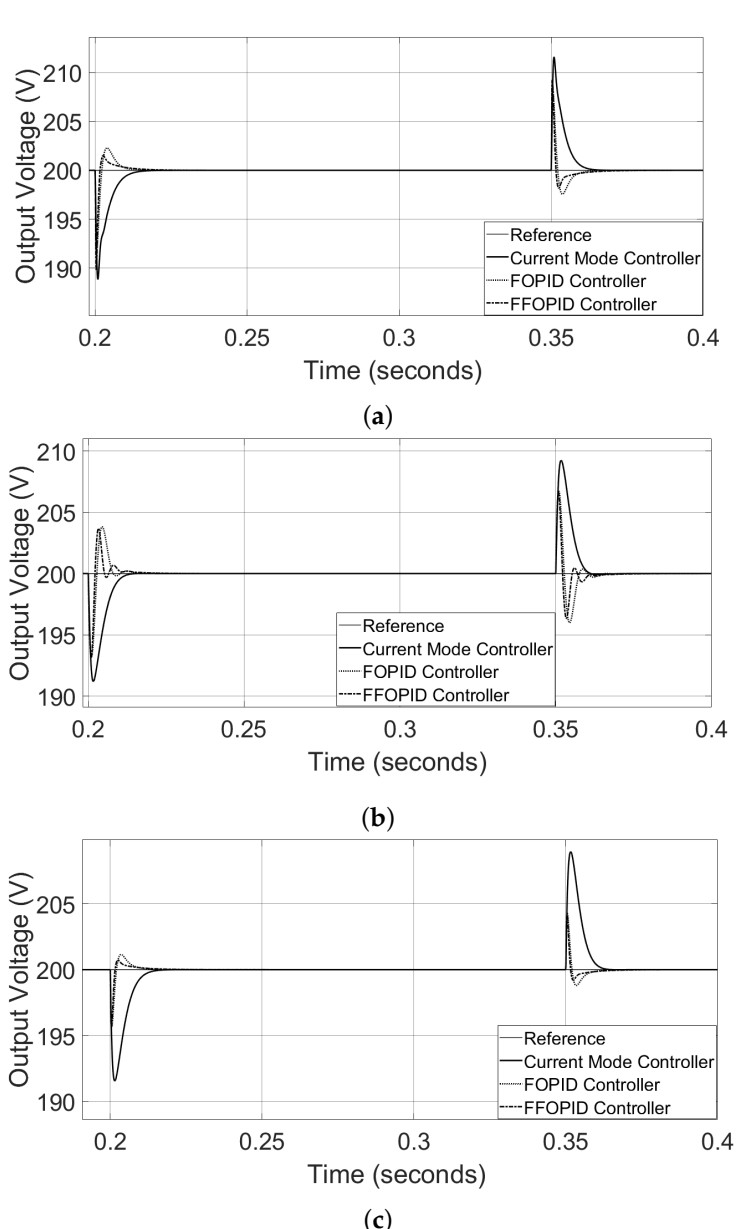

**Figure 12.** *Cont.*

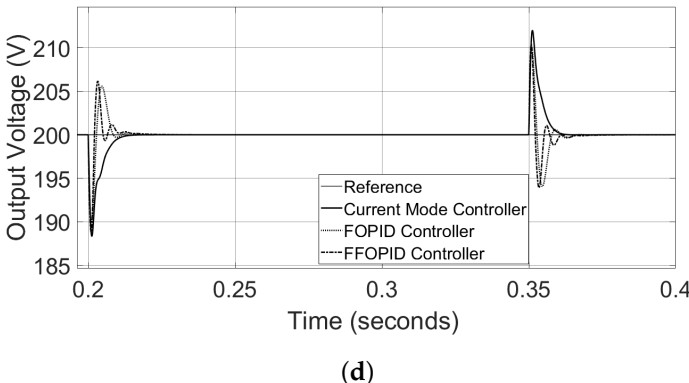

**(d)**

**Figure 12.** (**a**) Output voltage response for a capacitance startup deviation of −30%. (**b**) Output voltage response for a capacitance startup deviation of 30%. (**c**) Output voltage response for an inductance startup deviation of −50%. (**d**) Output voltage response for an inductance startup deviation of 50%.

Finally, the last case assesses the enhancement of the FFOPID against the FOPID under step control variable disturbances at the instant 0.1 s, without capacitance and inductance deviations. With the online updating of the gain parameters provided by the FLC, a relative improvement in amplitude of 19.81% of the FFOPID against the FOPID when a step control variable disturbance of −15% from the nominal Duty Cycle is applied can be seen in Figure 13a. In Figure 13b, a relative enhancement is found in the amplitude of 20.96% of the FFOPID against the FOPID when a step control variable disturbance of +15% from the nominal Duty Cycle is applied. By performance indexes comparison, besides ITAE and ITSE, the Integrated Absolute Error (IAE) and the Integrated Squared Error (ISE) were deployed to measure the behavior of the FFOPID and FOPID controllers during the control variable disturbance event of −15% from the nominal Duty Cycle. Table 10 shows these indexes when the time is measured from 0.1 to 0.13 s and sampled with $10^{-4}$ s.

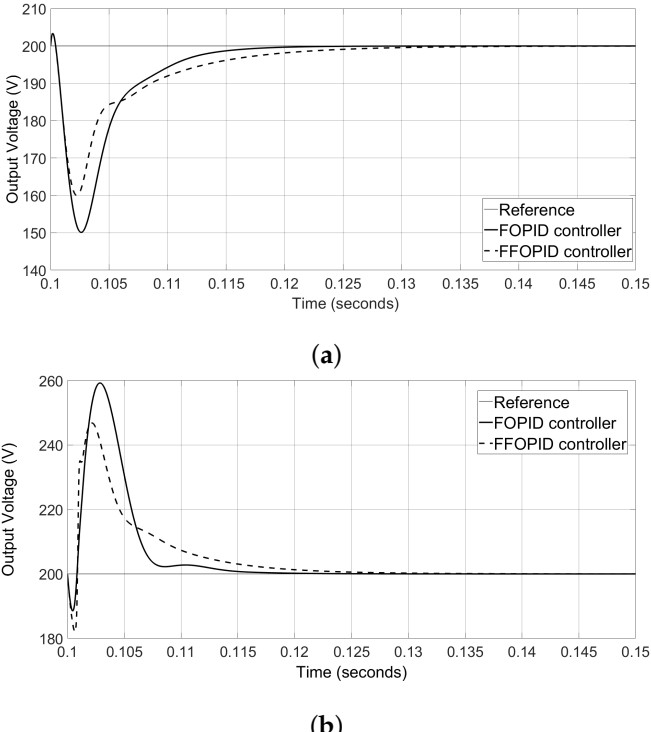

**Figure 13.** (**a**) Output voltage response for a step control variable disturbance of −15% from the nominal Duty Cycle. (**b**) Output voltage response for a step control variable disturbance of +15% from the nominal Duty Cycle.

**Table 10.** Performance indexes of FFOPID and FOPID controllers upon the disturbance of $-15\%$ from the nominal Duty Cycle.

| Index | Controller | |
| --- | --- | --- |
| | **FFOPID** | **FOPID** |
| IAE | 0.2298 | 0.2327 |
| ISE | 4.4599 | 6.9301 |
| ITAE | 0.0245 | 0.0244 |
| ITSE | 0.4631 | 0.7159 |

One can notice that the FFOPID controller is not more robust than the FOPID controller only with the ITAE measurement rule due to its lower settlement time. However, the others reveal the FFOPID controller superiority over the FOPID controller.

## 4. Discussion

The dynamical and phase margin specifications measured for the proposed FOPID were $\omega_{gc} = 2\pi240$ rad/s and $\phi_m = 59.6°$, respectively. The phase margin achieved in this manuscript is almost analogous to the FOPID designed in [17], showing that, with no direct assignment of phase margin specification, the proposed method matched robust phase margin against disturbances. Besides, it is remarkable that the amount of parameter variation analysis in this manuscript enhances the robustness study of FOPID for DC-DC Boost converters in the literature. However, the limitation of the proposed method regarding the one in [17] is its huge closed-loop transfer function size with a 30th order, yielding problems for hardware implementation such as higher memory requirements and physical space.

One of the key results of the FFOPID proposed is its robustness operation upon low power, i.e., full load to non-load, as seen in Figure 10. This feature is pertinent because even with a load as 10 K$\Omega$, the converter demonstrates a wide operational range along with high power regulation comparisons up to 50% above its power specification.

The zero initial voltage and its derivative for any range over derivative and integral fractional order of the proposed FOPID topology demonstrated the effectiveness of zero undershoot for the non-minimum phase DC-DC Boost converter when system standardization is pre-assumed. This feature allows the range of search in GA as related to $\mu$ and $\lambda$ for values above one, increasing the range possibility when compared to [14,16,17]. It is worth mentioning the regulation superiority of the proposed controller when simulating different startup reactive parameters along this manuscript while loads disturbances occurred, concluding the high robustness of the proposed FFOPID controller.

Finally, it is important to highlight that our work proposed a novel FOPID topology with higher robustness than the previous works in the literature for a wide operational range (power and components deviation). The disadvantages of the previous works are basically based on the usage of two sensors, no robustness analysis or a restricted range of optimal solutions, overshoots or poor settling times.

## 5. Conclusions

A GA-based FOPID controller tuning has been designed and applied to a DC-DC Boost converter. From the simulations studied, it is noticeable that the optimized controller parameters obtained by implementing the proposed algorithm with a probabilistic constraint in the weighted of ITAE and ITSE as performance indices have reached better controllability for load disturbance, and, greater setpoint tracking as regards the others due to parameters deviation of the system caused by the prolonged use of the equipment regarding controllers in voltage and current modes. It is also remarkable that the range of parameters deviation in this manuscript is higher than the standard deviation of $\pm20\%$, generally applied in power electronics designs, showing applicability for worst conditions.

In addition, the purpose of removing the voltage undershoots for all the ranges was achieved as seen in Figure 11, demonstrating the feasibility of the proposed FFOPID

controller. It is important to highlight the full range operation of the converter, i.e., full load and non-load with remarkable transients using only one voltage sensor (Figure 10). The reduction of sensors is in demand in both the academic world and industry. Finally, the FFOPID controller outperformed the FOPID controller in dynamic operation even upon $\pm 15\%$ deviation of the nominal duty cycle. Future research of this topic should be into the size reduction of the proposed FOPID using norm-2 and embedding it in an FPGA device to perform FPGA-in-the-loop simulations and applications in a prototype.

**Author Contributions:** Conceptualization, L.F.d.S.C.P. and E.B.; methodology, L.F.d.S.C.P., M.A.G.d.B., R.B.G. and E.B.; software, L.F.d.S.C.P.; validation, L.F.d.S.C.P. and E.B.; formal analysis, L.F.d.S.C.P. and R.B.G.; investigation, L.F.d.S.C.P., M.A.G.d.B and R.B.G.; writing—original draft preparation, L.F.d.S.C.P.; writing—review and editing, E.B., R.B.G. and M.A.G.d.B.; project administration, E.B. All authors have read and agreed to the published version of the manuscript.

**Funding:** This study was financed in part by the Coordenação de Aperfeiçoamento de Pessoal de Nível Superior—Brasil (CAPES)—Finance Code 001.

**Data Availability Statement:** Not applicable.

**Acknowledgments:** The authors want to thank the Federal University of Mato Grosso do Sul—UFMS and the Research and Development Project—P&D ANEEL. PD-06961-0010/2019.

**Conflicts of Interest:** The authors declare no conflict of interest.

## Abbreviations

The following abbreviations are used in this manuscript:

| | |
|---|---|
| FOPID | Fractional Order Proportional-Integral-Derivative |
| GA | Genetic Algorithm |
| ITAE | Integral of Time-Weighted Absolute Error |
| ITSE | Integral of Time-Weighted Square Error |
| ISE | Integrated Square Error |
| IAE | Integrated Absolute Error |
| QBGA | Queen Bee Assisted Genetic Algorithm |
| ABC | Artificial Bee Colony |
| FLC | Fuzzy Logic Controller |
| FFOPID | Fuzzy FOPID |
| CCM | Continuous Conduction Mode |
| FPGA | Field Programmable Gate Array |

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
