# Peer review of "A Robustness Analysis of a Fuzzy Fractional Order PID Controller Based on Genetic Algorithm for a DC-DC Boost Converter"

_electronics, doi:10.3390/electronics11121894_

Round 1
Reviewer 1 Report
Title: A Robustness Analysis of a Fuzzy Fractional Order PID Controller Based on Genetic Algorithm for a Boost DC-DC Converter
1) Why use fuzzy and GA algorithms? I think the motivation is not that strong. How about using MPC or PSO algorithms? The explanation should be given in the introduction.
2) As the authors said, comparisons among the proposed fuzzy logic FOPID controller with several conventional controllers were performed, but how to guarantee the fairness? Please give it out.
3) The reason for how to define the fuzzy membership and rule should be added. If I select other memberships or set other kinds of rules, will it affect the results?
4) In Figure 8, the conventional PI controller has a worse response speed. Please give more details about this. I personally think the PI parameters have not been finely tuned. This also can be found in Figures 9 (a) and (b) where the result with PI control shows more oscillations.
5) In Figure 11, why initial voltage is not stable and showing an increasing trend?
6) No stability analysis can be found in this study.
7) What is the novelty of the fuzzy and GA methods developed in this paper? Are they both the conventional ones just adopted for the power conversion here? If so, the contributions of the paper will be questioned.
Author Response
The authors would like to thank all the efforts done by Reviewer 1 to help improve our papers’ quality.

Reviewer 2 Report
I think the authors should report significant results in the abstract to promote the performance of their proposed idea.
About the literature review, some other papers should be added and each paper should clearly specify what is the proposed methodology, novelty and results with experimentation. At the end of related works, highlight in some lines what overall technical gaps are observed in existing works, that led to the design of the proposed approach. To better delineate the context and the different possible solutions, you can consider the following papers as references: "A Jxta Based Asynchronous Peer-to-Peer Implementation of Genetic Programming". J.Softw. 2006, 1, 12–23. and https://www.mdpi.com/2079-9292/10/18/2250.
The future scope of the methodology should be extended/highlighted. The limitations of this paper should be discussed. Accordingly, the future work of this paper can be drawn.
Author Response
The authors would like to thank all the efforts done by Reviewer 2 to help improve our papers’ quality.

Reviewer 3 Report
Review of electronics-1750017 entitled controller Based on Genetic Algorithm for a Boost DC-DC Converter, prepared by Luís Pereira , Edson Batista * , Moacyr A. G. De Brito , Ruben B. Godoy.
Paper presents the application of a Fractional Order PID (FOPID) controller to control a boost DC-DC converter with minimum over/undershoot. So the paper aim is actual. It suits to Electronics journal scope. Also, this paper has the potential to be cited because discusses the hot topic.
However, I have some recommendations that must be applied before final acceptance.
-> please extend the literature review with at least 3 publications from the 2022 year.
-> add a list of main abbreviations that are used in this paper.
-> for section "2. Materials and Methods" please add the adequate references that support the equations.
-> in Table 1. Specifications of Converter Prototype. please remember to use space between 8A.
-> in section 3 please add detailed information about SIMULINK of MATLAB versions and the device was used to perform research.
-> in section 3 please extend the discussion. Now only results are presented. Also please refer to the adequate literature.
-> add future research directions in section 4.
Author Response
The authors would like to thank all the efforts done by Reviewer 3 to help improve our papers’ quality.

Round 2
Reviewer 1 Report
Thanks for the response, I have no more questions.
Author Response
The authors are grateful for the reviewer's suggestions.
Reviewer 2 Report
The suggestions provided in the previous revision should be cosidered.
Highlight the importance of the proposed work in comparison with the existing techniques in the literature review discussion.
Author Response
The authors highlighted the importance of the work comparing with the literature review, as suggested by the reviewer.

Reviewer 3 Report
This paper has been significantly improved. Thus I recommend publishing it in its present form.
Author Response

(The authors gave the same response as above.)
